Evaluation study of effect of virtual care education on healthcare providers’ knowledge, confidence, and satisfaction

Clemens Megan megan.clemens@mun.ca
Boparai Josheil
Glynn Robert
White Gerry
Curran Vernon
Faculty of Medicine, Memorial University of Newfoundland , St. John’s, Newfoundland and Labrador , Canada
Prazeres Filipe
Electronic publication date: 2025 Nov 20
Publication date: 2025
Volume: 13
Electronic Location ID: e20414
Received 2025 Aug 14; Accepted 2025 Oct 28
Copyright: © 2025 Clemens et al.
Copyright year: 2025
Copyright holder: Clemens et al.
License: This is an open access article distributed under the terms of the Creative Commons Attribution License, which permits unrestricted use, distribution, reproduction and adaptation in any medium and for any purpose provided that it is properly attributed. For attribution, the original author(s), title, publication source (PeerJ) and either DOI or URL of the article must be cited.
License URL: https://creativecommons.org/licenses/by/4.0/

Keywords: Virtual care, Medical education, Continuing professional development, Telemedicine

Funding: The authors received no funding for this work.

==============================
Background

Virtual care can increase access to healthcare and improve provider efficiency; however, many healthcare providers lack formal education in virtual care delivery, including skills in virtual communication, physical examination adaptations, confidentiality, and billing procedures. This training gap can result in reduced confidence and suboptimal patient care. To address this, an asynchronous continuing professional development (CPD) module was developed. The objective of this study was to evaluate the module’s efficacy regarding satisfaction and changes in knowledge and confidence.

Methods

The module covered key topics such as virtual visit etiquette, technology troubleshooting, adapted physical examinations, documentation, and remuneration processes. Interactive features included embedded videos, knowledge-check quizzes, and reflective questions. A single-group pre-post quasi-experimental design was used to evaluate its impact. Data were collected via electronic surveys administered at three time points (before, during, and post-module). Surveys included multiple choice questions assessing objective knowledge, and Likert-scale questions assessing confidence levels in virtual care delivery. Open-ended short answer questions captured qualitative feedback. Quantitative data were analyzed using descriptive statistics and paired t-tests or Wilcoxon signed-rank tests where appropriate. Qualitative data were analyzed thematically to identify learner-reported strengths and areas for improvement.

Results

A total of nine to 22 learners responded at each time point. Respondents were heterogeneous, with most identifying as male (66.7%), general practitioners (55.6%), practicing in hospital settings (55.6%), and in communities of 2,000 to 10,000 people (55.6%). Learners reported high satisfaction with the module’s content relevance, navigation, and interactive components, but requested more interactive components (e.g., case-based learning). Statistically significant improvements were observed in confidence levels (n = 20–21; p < 0.001 to 0.009) and objective knowledge scores (n = 20–22; p = 0.046).

Conclusion

This evaluation study demonstrated that the asynchronous virtual care module had a statistically significant impact on objective knowledge and confidence, in addition to having positive satisfaction ratings. Limitations include the small sample size and lack of long-term follow-up to assess sustained practice change. However, these findings support the incorporation of asynchronous, virtual modules into CPD curricula to enhance provider competencies in virtual care delivery. Future directions include integrating additional case-based and specialty-specific content, as well as exploring the module’s scalability for other health professions to promote interprofessional virtual care training.

Introduction

Virtual care refers to the delivery of healthcare services using telecommunication technologies, facilitating healthcare from a distance (Curran, Hollett & Peddle, 2023a). Offering virtual care can make healthcare access more efficient for patients by reducing wait times (Appireddy et al., 2019; Rabinovitch et al., 2022; Young, Gupta & Palacios, 2019), reducing appointment duration without compromising patient satisfaction (Appireddy et al., 2019), and allowing patients to spend less time and money on travel (Welk, McArthur & Zorzi, 2022). The benefits of virtual care were further highlighted by the COVID-19 pandemic when in-person healthcare provision was restricted (Curran, Hollett & Peddle, 2023b; Heyworth et al., 2020).

However, it is also crucial to consider the healthcare provider’s perspective on offering virtual care services. Research has shown that healthcare providers prefer virtual care in certain contexts. For example, a survey found that physicians favored telemedicine for follow-up visits, medication refills, urgent care, behavioral health, dermatology visits, and chronic care management (Nies et al., 2021). Similar trends have been reported in the United Kingdom and Australia, where clinicians cite convenience and improved continuity of care as key benefits of virtual delivery models (Greenhalgh et al., 2020; Snoswell et al., 2020). Healthcare providers generally have a positive impression of virtual care (Kim et al., 2022), appreciating its convenience and efficiency (Li et al., 2022).

Despite the increasing relevance of virtual healthcare delivery, many healthcare providers receive little to no formal training in providing these services. This lack of training could explain the lower confidence that results in providers hesitating to provide virtual care services (Curran, Hollett & Peddle, 2023c), as they struggle to adapt their in-person clinical skills to meet the needs of a virtual environment (Teichert, 2016). This paucity in training begins early in medical education with medical learners having historically received insufficient training in virtual care techniques (Stovel et al., 2023; Strong, Dossett & Sandhu, 2020), contributing to providers’ downstream lack of confidence. Formal education that puts the theory of virtual care into action is crucial for healthcare providers to feel competent in offering this important service.

Virtual care delivery has become a widely adopted model of healthcare delivery in Canada and internationally, particularly following the COVID-19 pandemic (Canada Health Infoway, 2020; Webster, 2020), during which 71% of Canadian seniors reported having a virtual appointment, the highest proportion among surveyed countries (mean 39%; CIHI, 2022). Additionally, post-pandemic, 98% of primary care providers offer virtual care, most of which involves telephone appointments (CIHI, 2022), demonstrating a clear place for these services in Canadian healthcare delivery. As its use expands across diverse clinical contexts, it is essential to equip healthcare providers with appropriate training to ensure the delivery of safe, effective, and high-quality care. However, few studies have evaluated continuing professional development (CPD) training programs on virtual care for healthcare providers, and existing interventions vary widely in format, content, and assessment methods (Felker et al., 2022; Khan et al., 2021). Few programs integrate both objective knowledge testing and self-perceived confidence measures, and even fewer explicitly ground their design in established learning theory (Edirippulige & Armfield, 2016). Additionally, most published evaluations report only post-intervention satisfaction without a pre-post comparison, which limits inferences about actual learning gains (Waseh & Dicker, 2019).

The CPD module evaluated in the present study was designed to address these limitations by combining (a) theory-informed instructional design grounded in constructivism, (b) application-based case scenarios, and (c) structured pre-/post-assessment of both knowledge and confidence. Given these gaps, this study evaluates the effectiveness of a newly developed virtual care CPD module by examining learner satisfaction and changes in knowledge and confidence, providing one of the few theory-informed, pre–post analyses of virtual care education in a healthcare provider population.

Materials and Methods

Study design

This study employed a single-group pre-post quasi-experimental design, evaluating changes in knowledge and confidence before and after exposure to the CPD module.

Learning module

The Office of Professional & Educational Development (OPED), a part of Memorial University of Newfoundland’s Faculty of Medicine, hosts a number of online, asynchronous CPD learning modules. In February 2024, the OPED launched their first module on the topic of virtual care, titled Effective Virtual Care: Elevating Primary Care through Virtual Practice. The module’s design was informed by the constructivist learning theory which states that learners actively construct their understanding and knowledge through experience and reflection (von Glasersfeld, 1991). As a result, in designing the module, the goal was to help learners construct what they learn and understand through active participation (Rufii, 2015). To achieve this, learners were first given information on several topics, including an introduction to virtual care and related considerations of privacy/confidentiality, regulatory policies, and technical logistics; the integration of virtual care into an existing practice, including choosing the appropriate patients for this medium; and conducting a virtual appointment itself, such as adapting one’s clinical skills to this delivery method. The incorporation of a case study in the module helped to conceptualize the concepts in a meaningful healthcare context which aligns with constructivism (von Glasersfeld, 1991). To promote active engagement, learners completed pre- and post-test objective knowledge surveys. Throughout the module, there was a focus on providing external resources to enable learners to access further information on any particular topic. An example of the type of information provided, along with its format, is shown in Fig. 1. Additionally, a short case study was incorporated into the module.

Figure 1 Screen capture of the asynchronous virtual care CPD module interface.

Example view of asynchronous virtual care CPD module interface with embedded video, quiz elements, and navigation features.

Participants

The module was hosted on the MDcme.ca platform and was advertised to everyone with a registered account. This platform is primarily targeted to family medicine physicians; however, any healthcare provider, including but not limited to specialist physicians, residents, and nurses, may avail of the training courses. All registered MDcme.ca users were able to register and complete the module at no cost. At the beginning of the module, every participant was given the option to provide consent for their data to be collected for research purposes. The module was certified for 1.5 Mainpro+CPD credits with the College of Family Physicians of Canada.

The inclusion criteria were any registered MDcme.ca user who accessed the module and provided consent for their responses to be included in research. Exclusion criteria included duplicate entries and surveys without consent for research.

Due to the open-access nature of the module and voluntary participation in each evaluation component, the number of respondents varied across survey time points. Analyses were conducted on available case datasets rather than a single fixed cohort. Only participants with matched pre- and post-responses were included in paired analyses (confidence and MCQ knowledge), while all available responses were included in descriptive summaries of satisfaction and qualitative feedback.

Theoretical support

The evaluation is guided by the Kirkpatrick model, a widely used framework for assessing training programs (Alliger & Janak, 1989). Adapted for medical education, the modified model includes four levels: Learner satisfaction (Level 1), change in opinion (Level 2A), change in knowledge and skills (Level 2B), change in behavior (Level 3), and change in healthcare outcomes (Level 4) (Fig. 2). Based on the available evaluation data, this study focuses on levels 1, 2A, and 2B to assess early outcomes of the CPD module.

Figure 2 Modified Kirkpatrick model illustrating progressive levels of medical education outcomes.

Each level of the pyramid represents progressively deeper educational outcomes, from participant satisfaction (Level 1) to behavior and system-level changes (Level 4).

Data collection

There were a series of surveys asked across three time points: Before, during, and after the module (Table 1). Each of the surveys can be found in Appendix A. The survey items were developed by the research team based on core competencies outlined in virtual care literature and reviewed for content validity by faculty experts in CPD and virtual care. While the surveys were not formally validated prior to use, several satisfaction items reflect wording commonly used in accredited CPD evaluations, such as the College of Family Physicians of Canada’s requirement to assess perceived knowledge enhancement (College of Family Physicians of Canada, 2023). The rubric for short-answer questions was adapted from established approaches to scoring educational assessments (Popham, 1997; Moskal & Leydens, 2000). The pre- and post-test objective knowledge surveys were required to obtain the certification and CPD credits; however, there was not a minimum passing score. Each of the other evaluation points were voluntary and not required to complete the module.

Table 1 Content and timing of evaluation surveys administered in relation to the online module.

The four surveys used to evaluate the module, including their administration time points, question formats, and the specific constructs assessed (objective knowledge, confidence, and learner satisfaction).

Survey	Time point	Question types	Components evaluated	
Survey 1	Before module	6 MCQ	Objective knowledge	
6 Likert (1–4)	Confidence	
Survey 2	During module	8 short answer	Objective knowledge	
Survey 3	End of module	6 MCQ	Objective knowledge	
6 Likert (1–4)	Confidence	
Survey 4	End of module	10 Likert (1–5)	Learner satisfaction	
2 short answer	Learner satisfaction	
Note:

MCQ: Multiple choice question.

Survey 1

Survey 1 was delivered at the beginning of the module. Survey 1 contained 12 questions, six multiple choice questions (MCQs) pertaining to knowledge about virtual care, each with four answer options; and six questions about self-reported confidence in virtual care, each answered on a 4-point Likert scale (1 = not very confident to 4 = very confident).

Survey 2

During, but near the end, of the module, eight short answer questions were provided which assessed objective knowledge (Modified Kirkpatrick Model Level 2B). Each of them were related to a case study about virtual care from within the module.

Survey 3

Survey 3 was identical to Survey 1 (i.e., six MCQs which assessed objective knowledge and six Likert scale questions which assessed confidence) and delivered at the end of the module.

Survey 4

Survey 4 was an exit evaluation survey administered at the end of the online course. Learner satisfaction (Modified Kirkpatrick Model Level 1) was assessed quantitatively using four-point Likert scales (1 = not satisfied to 4 = very satisfied) through 10 questions: Two about course content, four about course navigation, and four about interactive components. There were also two short answer questions which assessed satisfaction qualitatively (What did you like about this course? How could we have improved this course?).

Survey 4 also contained six demographic questions. Each was answered in a categorical manner, for example “years of experience” had the options of 0–5 years, 6–10 years, 11–15 years, 16–20 years, or greater than 20 years.

Data analysis

Demographics were analyzed using descriptive statistics to report as both frequencies and percentages. Quantitative learner satisfaction was reported as both frequencies and percentages and categorized as positive (Likert scale 4 or 5), neutral (3), or negative (1 or 2). Qualitative learner satisfaction answers were reported as direct quotations due to their small number and brief nature.

Change in confidence was analyzed by calculating the difference between pre- and post-module scores for each question. Similarly, knowledge scores obtained from MCQs were calculated by summing the pre- and post-module scores separately. The normality of the score distributions were assessed using Shapiro-Wilk tests. As both variables violated normality assumptions, non-parametric Wilcoxon signed-rank tests were used to assess statistical significance. Results were reported as mean score differences with corresponding frequencies.

Short answer knowledge questions were independently evaluated by two reviewers (M.C. and J.K.B.) using a structured rubric in Microsoft Excel. Each response was scored out of 5 based on three predefined criteria: Completeness (40%), specificity (40%), and relevance (20%). This rubric aligns with best practices that recommend three to five evaluative dimensions (Popham, 1997) and reflects established approaches to holistic assessment (Brookhart, 1999), an emphasis on depth of response (Hubbard, Potts & Couch, 2017), and content-related validity (Moskal & Leydens, 2000). Final scores were calculated as the average between the two raters and reported using means and frequency distributions.

Ethics

Ethical approval was obtained from the Newfoundland and Labrador Health Research Ethics Board (reference #20250551). Written informed consent was obtained electronically; participants reviewed a written consent form and indicated agreement by selecting an electronic consent button prior to beginning the survey.

Results

Demographics

Between two and 22 participants completed each of the module and survey items. Participants represented a heterogenous group (Table 2). Just over half (55.6%) identified as general practitioners or family physicians, with the remainder including specialists (11.1%), nurse practitioners (11.1%), and other healthcare providers (22.2%). Practice settings were evenly distributed across solo (22.2%), group (22.2%), hospital-based (11.1%), combined group and hospital (22.2%), and other practice models (22.2%). Most respondents reported working in hospital-based environments (55.6%), typically within communities of 2,000–10,000 people (55.6%). Two-thirds of participants identified as male.

Table 2 Demographic characteristics of module respondents (n = 9).

Participant demographics, including profession, practice type and setting, community population size, and sex, with values reported as counts and percentages.

Characteristic	Category	n (%)	
Profession	General or family practitioner	5 (55.6%)	
Specialist	1 (11.1%)	
Nurse practitioner	1 (11.1%)	
Other	2 (22.2%)	
Practice type	Solo	2 (22.2%)	
Group	2 (22.2%)	
Hospital or institution	1 (11.1%)	
Group & hospital	2 (22.2%)	
Other	2 (22.2%)	
Practice setting	Hospital	5 (55.6%)	
Walk-in clinic	1 (11.1%)	
Private practice	2 (22.2%)	
Other	1 (11.1%)	
Community population	<2,000	2 (22.2%)	
2,000–10,000	5 (55.6%)	
10,000–20,000	1 (11.1%)	
>20,000	1 (11.1%)	
Sex	Male	6 (66.7%)	
Female	3 (33.3%)	

Satisfaction

Post-module likert scales

Participants were satisfied with the course, overall (Table 3). The course content evaluation questions each had 70% positive responses. Course navigation was the most favorable category, with responses ranging from 80% to 88.9% positive across the four questions. Finally, interactive components were positive (71.4% to 75%) except for one question about whether the “Ask the Expert” function (a feature where the learner can submit a question to a subject expert and receive an email response), which had only 42.9% positive responses and 57.1% neutral. Across all questions, there was only a single negative response.

Table 3 Participant ratings of post-module course satisfaction using a Likert scale.

Participant responses to course satisfaction prompts across three domains (course content, navigation, and interactive components) categorized as positive, neutral, or negative based on a 5-point Likert scale.

Prompt	n	Negative	Neutral	Positive	
Course content	
Addressed my learning needs	10	0 (0%)	3 (30%)	7 (70%)	
Enhanced my knowledge	10	1 (10%)	2 (20%)	7 (70%)	
Course navigation	
Instruction on use of and access to the course was helpful	9	0 (0%)	1 (11.1%)	8 (88.9%)	
The pages were well-organized	9	0 (0%)	1 (11.1%)	8 (88.9%)	
The pages were easy to navigate	9	0 (0%)	1 (11.1%)	8 (88.9%)	
I received adequate help with technical problems	5	0 (0%)	1 (20%)	4 (80%)	
Interactive components	
Participation in the discussion activities enhanced my understanding of the content	8	0 (0%)	2 (25%)	6 (75%)	
Being provided with the opportunity to communicate with peers was helpful	7	0 (0%)	2 (28.6%)	5 (71.4%)	
The discussion component was easy to use	8	0 (0%)	2 (25%)	6 (75%)	
The "Ask the Expert" option addressed my learning needs	7	0 (0%)	4 (57.1%)	3 (42.9%)	
Note:

Responses are categorized as positive (Likert scale 4 or 5), neutral (3), or negative (1 or 2).

Post-module short answer questions

The main positive responses to the short answer questions were regarding the information provided and the primary piece of feedback was integration of case-based learning and video delivery. The individual responses are presented in Table 4.

Table 4 Thematic summary of short-answer responses to post-module satisfaction questions.

Qualitative feedback on aspects of the course participants liked and suggestions for improvement, including representative verbatim comments.

Questions	Answers	
What did you like about this course?	“New information and how to manage our old habits”	
“Detailed information and well organized!”	
How could we have improved this course?	“Please, add some eduvational [sic] video clips”	
“Present case scenarios on how to access specialist easier to test systems and perform different case scenarios to prepare for virtual care.”	

Confidence

For the measures of confidence, the mean pre-module score (n = 21) was 16.48 out of a possible maximum of 24. The mean post-module score (n = 20) increased to 21.30. The distribution of scores at each time point can be viewed in Figs. 3 and 4, respectively.

Figure 3 Distribution of participant confidence scores prior to module completion (maximum score = 24; n = 21).

Histogram showing number of participants at each total confidence score out of 24 prior to starting the module; distribution skewed toward mid-range scores.

Figure 4 Distribution of participant confidence scores following module completion (maximum score = 24; n = 20).

Histogram of total confidence scores out of 24 following module completion; rightward shift compared to Fig. 3 indicates improved self-reported confidence.

A Wilcoxon signed-rank test demonstrated a statistically significant improvement for each of the following matched pre- and post-module questions: Question 7 (p < 0.001), 8 (p < 0.001), 9 (p = 0.003), 10 (p = 0.002), 11 (p = 0.002), and 12 (p < 0.001).

Knowledge

Within-module multiple choice questions

The first measure of knowledge was via a multiple-choice questionnaire at the beginning and end of the training module. The mean pretest score (n = 20) was 5.18 out of a possible maximum of 6. The mean post-test score (n = 22) increased to 5.73. The distribution of scores at each time point can be viewed in Figs. 5 and 6, respectively.

Figure 5 Distribution of participant knowledge scores on pre-module multiple-choice questions (maximum score = 6; n = 20).

Distribution of correct responses (out of 6) to knowledge-based MCQs before completing the module; most participants scored 6 but several responses in the lower range.

Figure 6 Distribution of participant knowledge scores on post-module multiple-choice questions (maximum score = 6; n = 22).

Histogram of correct knowledge scores out of 6 after completing the module; overall increase compared to pre-module scores in Fig. 5.

A Wilcoxon signed-rank test was performed, which demonstrated statistical significance (p = 0.046).

Within-module short answer questions

The within-module short answer questions showed a range of response accuracy (Table 5). Scores ranged from 0 to 5, with two questions having a mean score below 3: Question 2 (2.44) and Question 3 (3.94). The remaining questions had higher scores, with the highest being question 4 (4.56). Inter-rater reliability was assessed as the proportion of answers for which the coders’ scores were within one point of each other. Of the 94 answers evaluated, 65 met this criterion, yielding a percent agreement of 69.1%.

Table 5 Performance on within-module short-answer objective knowledge questions.

Participant scores for eight short-answer questions embedded within the module, reporting mean scores, score distributions, and the total possible score based on completeness, specificity, and relevance.

Question	n	Mean	Scores range	
<1	1–1.5	2–2.5	3–3.5	4–4.5	5	
Q1	17	3.53	0 (0%)	1 (5.9%)	2 (11.8%)	6 (35.3%)	6 (35.3%)	2 (11.8%)	
Q2	9	2.44	0 (0%)	3 (33.3%)	4 (44.4%)	0 (0%)	2 (22.2%)	0 (0%)	
Q3	9	2.94	0 (0%)	0 (0%)	3 (33.3%)	5 (55.6%)	1 (11.1%)	0 (0%)	
Q4	9	4.56	0 (0%)	0 (0%)	0 (0%)	0 (0%)	5 (55.6%)	4 (44.4%)	
Q5	10	3.35	0 (0%)	1 (10.0%)	2 (20.0%)	3 (30.0%)	3 (30.0%)	1 (10.0%)	
Q6	11	3.41	0 (0%)	0 (0%)	1 (9.1%)	8 (72.7%)	2 (18.2%)	0 (0%)	
Q7	10	3.15	0 (0%)	0 (0%)	5 (50.0%)	1 (10.0%)	4 (40.0%)	0 (0%)	
Q8	10	4.20	0 (0%)	0 (0%)	2 (20.0%)	1 (10.0%)	0 (0%)	7 (70.0%)	
Note:

Short answer questions were evaluated based on three criteria: Completeness (two points); specificity (two points), and relevance (one point), for a total possible score of 5.

Discussion

This study evaluated a newly developed CPD module on virtual care and found statistically significant improvements in learner confidence and objective knowledge, as well as high overall satisfaction. These findings suggest that such modules can be effective in addressing educational gaps in virtual care.

One key finding of this study was the high level of satisfaction. The majority of participants rated the course content, navigation, and interactive components favorably though some aspects of the interactive elements could be improved. Specifically, participants requested more interactive features, such as case-based learning and educational videos. Prior research has shown that the implementation of these elements can enhance student engagement and learning outcomes (Wang et al., 2025). Nunohara et al. (2020) found that videos and cases exert different influences on learners’ clinical decision-making, with video-case learning prompting greater attention to psychosocial aspects of care, whereas paper-based cases encouraged a stronger focus on biomedical elements. Given that virtual care demands a delicate balance between interpersonal abilities and medical knowledge, incorporating additional case-based activities and educational videos could improve learners’ capacity to integrate both domains, fostering a holistic and patient-centred virtual practice.

The preferences for increased interactivity reflect core principles of constructivist learning theory, which emphasizes that adult learners construct knowledge most effectively when actively engaged with realistic, problem-based scenarios (Vygotsky, 1978; Tam, 2000). This module’s design features structured content, learner autonomy, and embedded assessments, which align with best practices in instrumental design, particularly the use of multimedia learning principles and self-paced, modular structures to enhance cognitive processing and retention (Mayer, 2009). Expanding the module to include simulation- or video-based learning could further reinforce application of skills, consistent with emerging evidence in medication education (Wang et al., 2025). A recent randomized controlled trial demonstrated that interactive video-based case-based learning resulted in significant improvements in critical and systematic thinking abilities, as well as knowledge and satisfaction (Wang et al., 2025). This is particularly important in the context of virtual care training which must go beyond factual content to cultivate relational skills, empathy, and flexibility in practice (Nowell et al., 2025). Core framework principles—connection and interaction, compassion, empathy, care, vulnerability, client-centred focus, inclusivity, accessibility, and flexibility—map directly onto the gaps learners identified in our module (need for more case-based and interactive content) (Nowell et al., 2025). Incorporating interactive video cases and virtual simulations would therefore operationalize these principles by providing learners with safe, scaffolded practice in building rapport and interpreting nonverbal cues via video or phone.

Participants also demonstrated significant improvement in their post-test self-assessment scores, indicating increased confidence in applying their knowledge. Objective knowledge was also statistically significantly increased, measured via the post-test MCQ performance. These findings align with prior research, which shows that structured, interactive, and clinically relevant e-learning is more likely to improve learner satisfaction and knowledge retention (Cook et al., 2010; George et al., 2014). The intersections among knowledge improvements, confidence, and satisfaction increases demonstrate that module design features, such as interactivity and case-based learning, play a critical role in learner outcomes (Lawn, Zhi & Morello, 2017).

The rapid shift towards virtual care following the COVID-19 pandemic (Müller et al., 2021) left many providers unprepared to deliver quality virtual care (Fong, Baumbusch & Khan, 2024). As a result, the need for training modules like the one in our study is paramount. Curran et al. (2024) demonstrate that the delivery of CPD via e-learning can provide timely access to continuing professional education for health care providers who may be geographically dispersed. By demonstrating effectiveness across a heterogeneous group of learners, our study suggests that asynchronous CPD modules could be a scalable approach to virtual care training, particularly valuable for rural and remote providers who face barriers to attending in-person workshops (Calleja et al., 2022).

This study has a few key limitations. The small sample size evidently limits the generalizability of these findings; however, the population that was included is heterogeneous in several ways. Therefore, despite the small sample size, this study captured a variable group of participants and represented several perspectives. There are also limits to self-report measures; self-reported confidence gains may not fully reflect improvements in competency, a concern echoed in faculty development literature (Steinert et al., 2006). Additionally, the survey instruments were not previously validated, which may affect the generalizability of findings. However, the alignment with existing frameworks and expert review mitigates some concerns regarding reliability and validity. Finally, this evaluation study did not assess the higher modified Kirkpatrick model levels, behavior change (Level 3) or healthcare system outcomes (Level 4). Long-term follow-up could assess sustained practice change, which has been identified as a gap in CPD evaluation (Steinert et al., 2006). Therefore, the impact of this module on these more consequential outcomes is unknown at this time.

The findings of this study present several opportunities for fine-tuning future virtual care training opportunities. Firstly, interactive elements should be enhanced in order to address the neutral feedback from the survey. Additional case-based learning should also be incorporated to align with the qualitative feedback, an approach shown to improve application of knowledge (i.e., rather than just fact retention) in clinical settings (McLean, 2016; Wittich et al., 2010). Finally, assessment strategies should be expanded to include more objective knowledge measures, beyond MCQ and short answer questions, to better capture practical application of knowledge (Epstein, 2007).

Conclusions

This study is one of the first to evaluate a virtual care CPD module aligned with the Modified Kirkpatrick model. The findings demonstrated statistically significant improvements in objective knowledge and self-reported confidence, as well as overall positive satisfaction with content, navigation and interactive components. Participants valued structured content but expressed interest in more interactive and case-based learning. These insights highlight the potential of asynchronous e-learning for skill development in virtual care, while also identifying opportunities for improvement. Future versions of the module should prioritize the inclusion of case-based scenarios to improve practical relevance. Virtual care will continue to be a crucial part of healthcare provision, however, adequate training must be incorporated so this delivery modality is advantageous for both healthcare providers and patients.

Supplemental Information

Supplemental Information 1 Surveys administered at pre-, mid-, and post-module time points assessing knowledge, confidence, and satisfaction in virtual care delivery.

Supplemental Information 2 Answer key for multiple-choice assessment of knowledge and confidence administered pre- and post-module.

Supplemental Information 3 Results of multiple-choice assessment of knowledge and confidence administered pre- and post-module.

Supplemental Information 4 Dataset for pre- and post-module analysis of knowledge and confidence.

Supplemental Information 5 SPSS syntax used for analysis of pre- and post-module knowledge and confidence.

Supplemental Information 6 Output of analysis for pre- and post-module knowledge and confidence.

Supplemental Information 7 Short answer responses completed by participants within the module.

Supplemental Information 8 Scoring sheet with consensus results from two reviewers evaluating short answer responses.

Supplemental Information 9 Variable list for participant demographics and satisfaction survey about the virtual care module.

Supplemental Information 10 Results of participant demographics and satisfaction survey about the virtual care module.

Additional Information and Declarations

Competing Interests

The authors declare that they have no competing interests.

Author Contributions

Megan Clemens performed the experiments, analyzed the data, prepared figures and/or tables, authored or reviewed drafts of the article, and approved the final draft.

Josheil Boparai analyzed the data, authored or reviewed drafts of the article, and approved the final draft.

Robert Glynn conceived and designed the experiments, authored or reviewed drafts of the article, and approved the final draft.

Gerry White conceived and designed the experiments, authored or reviewed drafts of the article, and approved the final draft.

Vernon Curran conceived and designed the experiments, authored or reviewed drafts of the article, and approved the final draft.

Human Ethics

The following information was supplied relating to ethical approvals (i.e., approving body and any reference numbers):

Ethical approval was obtained from the Newfoundland and Labrador Health Research Ethics Board (reference #20250551).

Data Availability

The following information was supplied regarding data availability:

The raw data are available in the Supplemental Files.

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
