# Peer review of "Evaluation study of effect of virtual care education on healthcare providers’ knowledge, confidence, and satisfaction"

_PeerJ, doi:10.7717/peerj.20414_

## Round 0.1 · original submission · Minor Revisions

· Academic Editor

Minor Revisions

I have read the feedback from each of the reviewers and recommend that you revise your manuscript in line with their comments and the PeerJ editorial criteria: https://peerj.com/about/editorial-criteria/.

Reviewer 1 ·

Basic reporting

Reporting seems ok.

Experimental design

Design not explicitly stated

Validity of the findings

Findings well presented

Additional comments

Line 89-93: What makes your CPD module unique? I think you need to provide more context on the current state of the literature regarding trainings, what authors have learned from them, and the exact gaps this study aims to fill. I see you provide details about the learning module in the methods, but a better context in the introduction would strengthen what has been provided about the program.

You did not explicitly state the study design. It looks like it could be implied that it is a pre-posttest quasi experimental study. Kindly indicate the design.

Line 188-193: I think this is good, but refrain from the choppy sentences.

Your discussion is not detailed and focuses only on two parts of the results. From line 244-249, you state that your findings align with previous research, and that is not very informative. What are some future implications. You need to expand your discussion.

Reviewer 2 ·

Basic reporting

The background is only skewed towards the context and doesn’t provide an interaction with literature from other contexts.

Experimental design

There is no information provided about sample size, participant characteristics, inclusion/exclusion criteria, or recruitment process, which limits reproducibility and understanding of generalizability.

The description of thematic analysis lacks detail about the coding process, the number of coders, inter-rater reliability, or the use of qualitative software. Furthermore, qualitative data is presented in the abstract but not in the document.

Participant characteristics are crucial; furthermore, they also include the inclusion and exclusion criteria

The criteria for choosing between paired t-tests and Wilcoxon tests are not specified.

Validity of the findings

There is no information provided about sample size, participant characteristics, inclusion/exclusion criteria, or recruitment process, which limits reproducibility and understanding of generalizability.

The total number of respondent need to be provided, and each number needs to be identified across each of the respondents

The discussion would benefit from development and further discussion on findings and intersections between concepts.

A clear depiction of whether piloting was done, validated, or adapted from existing tools raises questions about reliability and validity.

Additional comments

Kindly extend the discussion to support the interconnecting concepts within your study. Further develop these ideas to ensure a robust discussion.

Annotated reviews are not available for download in order to protect the identity of reviewers who chose to remain anonymous.

---

## Round 0.2 · accepted · Accept

· Academic Editor

Accept

I have reviewed the revised manuscript and confirm that the authors have satisfactorily addressed all of the reviewers’ comments. In my opinion, the manuscript is ready for publication.

Reviewer 1 ·

Basic reporting

No comment

Experimental design

Comment addressed.

Validity of the findings

No comment

Additional comments

All comments addressed.